# Generating Paraphrase Using Simulated Annealing for Citation Sentences

**Ridwan Ilyas \*, Masayu Leylia Khodra, Rinaldi Munir, Rila Mandala and Dwi Hendratmo Widyantoro**

School of Electrical Engineering and Informatics, Bandung Institute of Technology, Bandung 40312, Indonesia
\* Correspondence: rdwnilyas@gmail.com

**Abstract:** The paraphrase generator for citation sentences is used to produce several sentence alternatives to avoid plagiarism. Furthermore, the generation results need to pay attention to semantic similarity and lexical divergence standards. This study proposed the StoPGEN model as an algorithm for generating citation paraphrase sentences with stochastic output. The generation process is guided by an objective function using a simulated annealing algorithm to maintain the properties of semantic similarity and lexical divergence. The objective function is created by combining the two factors that maintain these properties. This study combined METEOR and PINC Scores in a linear weighting function that can be adjusted for its value tendency in one of the matrix functions. The dataset of citation sentences that had been labeled with paraphrases was used to test StoPGEN and other models for comparison. The StoPGEN model, with the citation sentences dataset, produced a BLEU score of 55.37, outperforming the bidirectional LSTM method with a value of 28.93. StoPGEN was also tested using Quora data by changing the language source in the architecture section resulting in a BLEU score of 22.37, outperforming UPSA 18.21. In addition, the qualitative evaluation results of the citation sentence generation based on respondents obtained an acceptance value of 50.80.

**Keywords:** citation sentences; paraphrase generator; simulated annealing

## 1. Introduction

Paraphrase generation produces new text from the input with different wording but the same information [1]. The generation machine aims to create sentences with high lexical divergence and maintain semantic similarity. Furthermore, the generator is often equated with a machine translation, but the input and output are sentences in the same language [2]. The generation of a paraphrase needs to consider several criteria. Moreover, sentences in scientific papers are usually argumentative [3], where one statement is bound in context with another, either in a causal paragraph or vice versa. The resulting new sentence should not have plagiarism characteristics [4]. Scientific papers contain many equivalent or multilevel compound sentences; hence, the output form is more complex [5]. In the domain of scientific papers, paraphrasing can be found in several events [6], such as:

1. The abstract is a paraphrase of the sentence in the body of the paper
2. The introductory part has a paraphrase equivalent to the methodology section
3. The conclusion has a paraphrase equivalent to the experimental section
4. Definition sentences have paraphrase equivalents with others that define the same construct
5. The citation sentence that quotes the same paper is a paraphrase.

This study used the potential of the citation sentence as the paraphrase collection, which can be obtained from various papers. The citation sentence was selected because it is often considered a part that increases a paper's plagiarism value.

It also has many potentials for paraphrasing when collected. Therefore, the dataset was collected from open-source Computing Language Papers (ACL Anthology). Citation sentences have several functions, including citing weaknesses, contrasts, methods, and data similarities, as well as problem bases or neutral ones [7]. The citation sentence used in this study is limited to only one of the citation targets. This was carried out to limit the context of the purpose of each delivery in the scientific argument on sentences.

Inspired by the *Unsupervised Paraphrasing of Simulated Annealing* [8], a generate and test model architecture was developed with the same algorithm but different objective functions and strategies. Furthermore, this study combined two matrix functions, namely METEOR [9] and PINC Score [10], to capture semantic similarities and lexical differences. The two matrix functions were combined in a linear weighted function [11], which can be adjusted to the tendency of its value. The language source that makes a substitute or addition successor to the input sentence is built with word embedding [12]. The sentence candidate selection strategy uses the n-gram language model [13].

Approaches for rule-based paraphrase generation are based on hand-crafted and automatically collected paraphrase rules. These rules were mostly hand-crafted in the early works [14]. Because of the enormous manual work required, some researchers have attempted to collect paraphrases automatically [15]. Unfortunately, because of the limitations of the extraction methods, long and complex additional patterns have been generated, affecting performances.

Thesaurus-based approaches start by extracting all synonyms for the words to be substituted from a thesaurus. The best choice is then selected according to the context of the source phrase [16]. Although simple and effective, the diversity of the generated paraphrases tends to limit this strategy.

Seq2Seq models were initially used for paraphrase generation with recent advances in neural networks, particularly the sequence-to-sequence architecture [17]. Convolutional neural networks (CNNs) have also been used to build seq2seq models since they have fewer parameters, and so train faster [18]. The Transformer [19] model has shown attempting to cut performance on a variety of text generation tasks. A transformer was developed for the seq2seq model because of the Transformer's improved capacity to capture long-term dependencies in sentences [20].

The dataset of citation sentences was tested using other methods with a supervised approach, such as LSTM, a type of recurrent neural network that can be used for paraphrase generation by learning to capture long-term dependencies in input text sequences [21]. Bi-LSTM for paraphrase generator is a neural network model that utilizes bidirectional processing of input sequences to generate paraphrases with a focus on capturing contextual information [22]. Transformer is a neural network architecture for natural language processing that uses self-attention mechanisms to encode and generate paraphrased sentences. Furthermore, this study used the unsupervised method by modifying UPSA [8] to compare the ability to produce paraphrased sentences. Apart from the citation sentence data, the paraphrased data from the questions in Quora [23] and Twitter [24] showed that the architecture built could be used for other domains. The results of StoP-GEN generation were compared with UPSA. Variant autoencoder is a neural network architecture that can be used for generating paraphrases by learning a compressed representation of the input text and then using it to generate a new text [25]. LagVAE is the improvement of the variant autoencoder [26]. CGHM (Concept-Phrase Hypergraph Model) is a model that generates paraphrases by leveraging semantic concepts and syntactic information to build a hypergraph representation of the original sentence [23].

The method of building the corpus and paraphrase generation has been widely developed for social media and news. However, the existing paraphrase generation method cannot be directly adapted to produce new sentences in scientific papers. Paraphrase generation is very dependent on the available language resources. The previously developed method was unable to provide alternative generation, so it requires a stochastic generation method.

This study contributed to producing paraphrase generation methods for citation sentences from scientific papers. This study focuses on developing the architecture of generating methods. Paraphrase detection formulas are used as objective functions. The method developed has a stochastic output and produces a different alternative output but the conference to the objective function. The proposed method has been tested in several corpora.

## 2. Related Work

### 2.1. Corpus Construction

The construction of the corpus paraphrase is known as paraphrase extraction. This is a task to generate the collection of paraphrased sentence pairs from large documents [27]. The extraction result can be a collection of words or phrases, such as PPDB [26], which uses two-language pivoting. It can also be the paraphrased sentence pair, such as MSRP [28], which is obtained from a news collection using a supervised approach. Other corpora, such as PIT [29], were compiled from tweets using the similarity object delivery approach. Each text unit and domain have unique characteristics because of its specific information purpose. State or the art of constructing a paraphrase corpus can be seen in Table 1.

**Table 1.** Paraphrase corpus state of the art.

| No | Paper | Year | Name | Domain | Technique |
|----|-------|------|------|--------|-----------|
| 1 | Ganitkevitch et. al. [27] | 2013 | PPDB | Free | Pivoting |
| 2 | Pavlick et. al. [30] | 2013 | PPDB 2.0 | Free | Pivoting |
| 3 | Dolan et. al. [28] | 2005 | MSRP | Free | SVM |
| 4 | Xu et. al. [29] | 2014 | PIT | Twitter | Multi-instance learning |

It is necessary to observe the authors' characteristics in conveying information when extracting paraphrases from scientific paper sources. Authors of scientific papers write information using three approaches, namely paraphrasing, summarizing, and translating [31]. Abstract sentences with body parts can be collected to build a paraphrase corpus [5]. However, citation sentences have the greatest potential to build a paraphrase corpus from these papers [32]. The construction of the citation paraphrase corpus in this study is a small contribution to paraphrase generation research.

### 2.2. Objective Function

The generation model built with the generate and test model requires an objective function to guide the generation process. In paraphrasing, the objective function is a formula to measure the paraphrase value of two pairs of sentences (usually a value between 0 to 1). Studies in this section are usually grouped in the task text similarity measurement.

Paraphrasing is a task that is very similar to machine translation; therefore, the evaluation approach of the translation can be used for paraphrasing. Furthermore, evaluation techniques, such as NIST [33], BLEU [34], and WMT [35], can be combined into a formula to assess the results of paraphrase generation evaluated based on the available data [36].

The Term Frequency Kullback–Leibler Divergence (TF-KLD) data representation is the best technique for measuring paraphrases in the MSRP dataset [37]. Prior to the classification, the matrix is converted into a latent representation with TF-KLD features, and the SVM algorithm is subsequently used for classification. The evaluation was carried out by comparing the standard TF-IDF resulting in an accuracy of 80.4% and an F1 Score of 85.9%.

Another approach for measuring the paraphrase output is the use of deep learning to build a sentence representation and simply compare it in vector form [32]. Apart from the neural network architecture, a Convolution Neural Network (CNN) model, which consists of composition layers, decomposition, and proximity estimation, can be used to

measure the paraphrase generation results [38]. Text representation with word embedding is often used when a deep learning approach is applied.

A model is developed to measure paraphrase in the domain of scientific papers. The Siamese neural network architecture is used to study the similarity and dissimilarity based on corpus labeled true and false for sentences from scientific papers [39] with an accuracy rate of 64%. Furthermore, the SVM model can be developed by engineering word features, such as Euclidean distance, cosine similarity, and sentence length [40], with an accuracy rate of 61.9%. Both studies utilized a learning-based approach and were strongly influenced by the quality of the corpus used.

In this study, the objective function was built based on semantic similarity and lexical divergence. To combine the two, a formula that can configure the tendency to one aspect was built. The objective function formation is explained in the experiment section.

### 2.3. Paraphrase Generator

Paraphrase generation is a task to generate new sentences from the input. Furthermore, various language resources are needed in this process. The general approach of generating paraphrases uses a machine translation set to produce sentences in the same language [2].

Paraphrase generation can be found in various domains, such as news [41], where the generator can be used to package news content or form variations of headlines [42]. It can also be found in social media domains such as Twitter [29]. Paraphrase generation in these various domains aims to produce semantic similarity, compression, concatenation, and sentence simplification [43].

The sequence-to-sequence learning is a technique developed with a deep approach to paraphrase generation [44]. The main construction of this model is the Recurrent Neural Network (RNN) or Long-sort Term Memory (LSTM) units. The deep learning approach was developed with the Transformer and inspired the use of this technique in paraphrase generation [45].

### 2.4. Simulated Annealing

Simulated Annealing (SA) is an effective algorithm in the solution search on a very large dimension space [46]. The advantage of the algorithm is its ability to avoid the local maximum of the optimization function. Furthermore, the algorithm is inspired by heavy industrial processing that utilizes the lowering of an object's temperature and manipulates it to the desired shape. The temperature drop factor determines the fault tolerance in the search process of the solution space. The error is acceptable when the temperature is still high and less likely to be accepted towards the end of the temperature drop.

In the sentence generation case, it can be stated that $\chi$ is a very large sentence dimension space, and $f(x)$ is the objective function of generating new sentences. The main target of Simulated Annealing is to determine the sentence x with the maximum value $f(x)$. There is a generation step in every search, which can be called $t$, while the sentence generated can be referred to as $x_t$. Simulated Annealing will select $x_{t+1}$, which has undergone a change from $x_t$ as the current step when the f value is greater. Otherwise, tolerance will be calculated based on the probability value $e^{\frac{f(x_t)-f(x_{t+1})}{T}}$ controlled by $T$ (temperature), which decreases at each step, resulting in 0 in the last step. For example, in $x_{t+1}$ a smaller $f$ value is obtained, and $x_{t+1}$ can be accepted as a new step when the result of generating a random number $r < e^{\frac{f(x_t)-f(x_{t+1})}{T}}$. Simulated Annealing is inspired by the metallurgical process of cooling materials. At the beginning of the search, the temperature $T$ is usually very high and allows $x_{t+1}$ to be accepted even though the value of $f$ is smaller. Theoretically, this can avoid the local maximum's optimization function and guarantee the global maximum's achievement [47].

### 3. Methods

#### 3.1. Generator Model

The process of forming new sentences is developed following the generate and test model mechanism. The use of the simulated annealing algorithm requires the changing instances process. Furthermore, input sentences are considered instances that are processed into different sentence forms. The sentence changes are carried out at the lexical level. In addition, there are three factors to consider when forming new sentences, namely action (type of shape change), language model (change reference source), and sentence change strategy.

In one of the generation processes, an object is selected with a choice of substitution, insertion, and deletion actions. The replacement process is carried out by selecting equivalent words from the language model, while the addition of objects is conducted by inserting a new object based on the selection in the model. The substitute word is selected based on the neighbor context in the input sentence; therefore, paying attention to the word before and after is necessary. Meanwhile, the deletion process removes the selected object from the sentence. The architecture of the method for paraphrase generation using simulated annealing is illustrated in Figure 1.

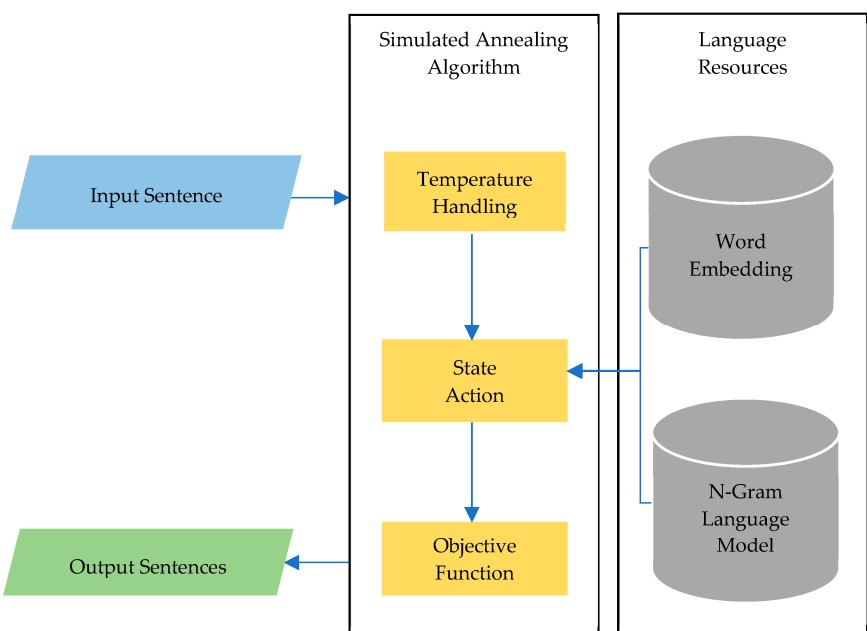

**Figure 1.** Method architecture for paraphrase generator.

Editing steps is a strategy for the sequence of action to be carried out. It can be selected in two ways, namely selecting, randomly and sequentially, from the beginning to the end of the sentence. Furthermore, the editing steps will determine how to stop the sentence generation process. When using a random scheme, the termination process entirely depends on the temperature setting of the simulated annealing algorithm. However, the process stops depending on the number of words in the sentence when conducted sequentially. With this strategy, the temperature is still used, and only the reduction process is adjusted to the number of lexical targets processed. An example of changing sentences in the generation process is shown in Figure 2.

| | Input | *The model was implemented using TensorFlow (abadi et al.,2015).* |
|---|---|---|
| **Substitution** | Target<br>Successor<br>New<br>sentence | *model*<br>*design*<br>*The design was implemented using TensorFlow (abadi et al.,2015).* |
| **Insertion** | Target<br>Accessory<br>New<br>sentence | implemented<br>*by*<br>*The design was implemented using TensorFlow (abadi et al.,2015).* |
| **Deletion** | Target<br>New<br>sentence | *using*<br>*The design was implemented using TensorFlow (abadi et al.,2015).* |

**Figure 2.** Example of the changing process of the sentence form based on action.

A successor is a substitute or additional text unit that will be included in a sentence. The source used to obtain the successor was obtained from the language model with the word embedding approach. This language model was created using word2vec [48] with a corpus as a source for scientific papers collection. Word2vec has two models, namely CBOW and Skip-Gram. CBOW allows performing successor search processes for substitution and insertion actions. Meanwhile, the skip-gram was not used because it could not find the successor for the insertion action. The cosine similarity formula (1) is used to determine the successor in the substitution action by comparing the target vector with the candidates in the word2vec dictionary. Meanwhile, the probability vector formula (2) is used to determine the successor by comparing the neighbor word vectors from the target position with the candidates in the word2vec dictionary.

The formula for calculating similarity:

$$Sim(A,B) \ = \ \frac{\sum_{i=1}^{n} A_i B_i}{\sqrt{\sum_{i=1}^{n} A_i^2} \sqrt{\sum_{i=1}^{n} B_i^2}} \tag{1}$$

The formula for calculating vector probability:

$$Prob \ = \ \sqrt{n \, \sigma^2 + {}^1/_n} \tag{2}$$

where $\sigma^2$: the variance of the elements. $n$: number the elements

The substitution and insertion actions make it possible to produce several new sentences because the results of the candidate selection allow more than one successor. Therefore, the sentences formed from all possible successors can be very diverse. In the simulated annealing algorithm, a state can only be filled by a sentence; hence, a selection strategy is required. The n-gram language model [13] is used to measure the word object arrangement probability of all possible sentences that can be formed. Function (3) assesses the new sentence structure when the successor is included.

The formula for the n-gram probability language model:

$$pLM(w_n|w_{n-N+1:n-1}) \ = \ \frac{C(w_{n-N+1:n-1} \, w_n)}{C(w_{n-N+1:n-1})} \tag{3}$$

where

$pLM(w_n|w_{n-N+1:n-1})$    : represents the probability of the word $w_n$ given the previous $N-1$ words $w_{n-(N-1)}, \dots, w_{n-1}$

| | |
|---|---|
| $C(w_{n-N+1:n-1}\, w_n)$ | : represents the count of the N-gram $w_{n-(N-1)}, ..., w_n$ in the training corpus |
| $C(w_{n-N+1:n-1})$ | : represents the count of the $N-1$ gram $w_{n-(N-1)}, ..., w_{n-1}$ in the training corpus |

The probability of choosing an action was analyzed in the experiment to see its influence on the generation outcome. Furthermore, multiple combinations of odds were allowed from three available actions; substitution, insertion, and deletion. For this reason, an action opportunity distribution scenario was prepared, as shown in Table 2.

**Table 2.** Action factor probability scenarios.

| Action Factor | Substitution | Insertion | Deletion |
|:---:|:---:|:---:|:---:|
| A | 90% | 5% | 5% |
| B | 80% | 10% | 10% |
| C | 70% | 15% | 15% |
| D | 60% | 20% | 20% |
| E | 50% | 25% | 25% |
| F | 40% | 30% | 30% |
| G | Uniform | | |

### 3.2. Objective Function

Simulated annealing works by optimizing the objective function. In the case of paraphrase generation, an objective function that evaluates each state based on semantic similarity and lexical divergence is required. Therefore, this study utilized two calculations of these aspects in one formula. The formula used is weighted linear (4) inspired by Maximal Marginal Relevance (MMR), which selected the information based on two contradictory parameters, one strengthening and the other weakening [11].

$$Weighted\ Liner = \alpha * P1 + (1 - \alpha) * P2 \tag{4}$$

where

$P1 = parameter\ value\ 1$
$P2 = parameter\ value\ 2$
$\alpha = multiplier\ constanta\ (0 - 1)$

Two calculation matrices are needed to fill in the parameter values in the formula, each representing semantic similarities and lexical differences. The METEOR was selected to obtain the semantic similarity value [9], and PINC Score [10] to obtain the lexical divergence. The two calculated functions fill in the two parameters in function (2).

METEOR [34] is a basic parameter used to measure semantic similarity. Furthermore, it is commonly used to assess the translated text quality. Compared to other measuring tools, it has the ability to calculate the semantic similarity value from different lexical items because it is equipped with a lexical similarity dictionary. In machine translation, METEOR's measurement results are closer to human judgment than other measuring instruments such as BLEU.

METEOR works by calculating the Precision (*P*) value, which is the ratio of the appropriate number of N-Grams in all translation results, and the Recall (*R*), which is the ratio value of the appropriate n-gram number in the translation results to the reference sentence. Furthermore, *FMeans* (5) is also calculated, namely the harmonic-mean value, which prioritizes the recall effect.

$$FMeans = \frac{10PR}{R + 9P} \tag{5}$$

Apart from the calculation of *FMeans*, METEOR considers the penalty value (6), which is the least number of *Chunks* in the phrase related to the reference, divided by the

corresponding N-Gram number. The penalty reduces the *FMean* value to better consider the appropriate *Chunk*.

$$Penalty = 0.5 * \left( \frac{\#chunk}{\#unigrams\_match} \right) \tag{6}$$

The final METEOR value is calculated by the formula (7):

$$MS = FMeans * (1 - Penalty) \tag{7}$$

Pinc Score [10] was used because of the need for a formula to calculate the lexical divergence between the input and generated sentences. The score (8) was used to complete the parameters of the weighted linear function. Furthermore, it calculates the number of words that do not intersect from two text inputs with N-Grams and can be adjusted as needed. The score is obtained by subtracting the number 1 from the suitability value of the two sentence units.

$$PS\,(O,S) = \frac{1}{N}\sum_{n=1}^{N} 1 - \frac{|ngram_o - ngram_s|}{|ngram_s|} \tag{8}$$

where

$O = system\ output\ sentence$
$S = reference\ sentence$
$ngram = number\ of\ matching\ words$
$N = the\ sensitivity\ value$

This study proposed the PScore, a weighted linear formula that uses METEOR and PINC Scores as two inverse parameters. A multiplier constant of 0.9 was obtained for METEOR. The PINC Score had a constant of 0.1. This value was selected based on the measurement of pre-experimental results. The weight used is unbalanced to emphasize one aspect. In the formula used, the meteor gets almost absolute value because the results obtained can prioritize semantic similarity and a little lexical divergence.

$$Pscore = 0.9 * MS + (1 - 0.9)\,PS \tag{9}$$

StoPGEN was proposed as a method for generating paraphrase sentences based on the simulated annealing algorithm and developed with three types of architecture. The first uses a random target word selection scheme as done by UPSA. The second uses a sequential selection scheme from the beginning of the sentence to the end. Meanwhile, the third uses a sequential selection scheme with the probabilistic language model to select successors. The third architecture with complete functionality is described in Algorithm 1.

---

**Algorithm 1: Simulated Annealing for Sentence Generator**

**FUNCTION** StoPGEN(x [0…n]: Array of token) return S [0…n]: Array of token
 {function of sentence generator using simulated annealing}
**DECLARATION**
 T, initial temperature
 ΔT, temperature drop
 s, current state {state is array of token}
 Pscore, objective function
 pLM (), language model probability function {return score}
 most_similar (), get most similar token from word index
 Neighbors (), get most similar token by neighbors from word index
**ALGORITHM**
 T ← T$_0$
 Sk ← x
 **while** T > 0 **and** n **do**

```
action ← get_random_action ()
if action = subtitution do
    sk + 1[n] = max (pLM (most_similar (sk + 1[n])))
else if action = insertion do
    sk + 1[n] = max (pLM (neighbors ([sk + 1[n − 2], sk + 1[n − 1], sk + 1[n + 1], sk + 1[n + 2]])))
else
    remove(sk + 1[n])
    ΔE ← Pscore (sk + 1)−Pscore (sk)
    if min (1, e−ΔE/T) >= rand (0,1) then
        sk ← sk + 1
    end if
        T ← T − ΔT
    end while
return sk {new array of token}
```

### 3.3. Dataset

Paraphrasing citation pairs were obtained from papers on the Association of Computational Linguistics (ACL) website (https://aclanthology.org/, accessed on 01 August 2019). The construction of the dataset runs in two stages, first, the candidate corpus development, and second, the labeling of the corpus sentences by the annotator. The construction of a candidate corpus is the extracting process from all collected papers to select citation sentences. Each sentence is processed by clustering technique based on the text features and the target citation similarity. The details of the parallel corpus candidate development process were published [6].

The following steps are taken in the process of getting a corpus:

1.  Sentences from scientific papers will be collected based on their function as citation sentences, abstracts, and content. This process uses the Dr. Inventor Framework [49].
2.  From the extraction results, the sentences selected white only have one citation target. The sentences that have two or more citation target was ignored.
3.  After the selection is made, the clustering process is carried out. Clustering sentence citations is the process of grouping sentences to get candidate pairs of sentences that have the same meaning. The clustering of citation sentences uses the K-Means algorithm [50] with Jaccard similarity, bigram representation, and TF-IDF.
4.  After the clustering process, each sentence in a cluster is paired with one another as a corpus candidate for the labeling stage.

Labeling is the last stage of producing a parallel corpus. Five annotators were used to label paraphrased or non-paraphrased. The number of data generated from the annotation process was 5720 pairs of sentences, with 4975 paraphrases and 745 non-paraphrases.

This study also utilized the Quora dataset in the question pair form, as used by UPSA [8], with CGHM as the setting of the dataset [23]. Furthermore, paraphrased sentence datasets from Twitter social media were also used [24]. Both of these datasets are used to test the performance of the architectures that are created and compared with other methods such as UPSA [8], Variant Auto Encoder [25], Lagging Variant Auto Encoder [51], and CGHM [23].

## 4. Experiment

### 4.1. Dataset Evaluation

The Fleiss kappa formula is commonly used to measure inter-annotator agreement when multiple annotators provide judgments on a categorical item. To evaluate the paraphrase dataset, multiple annotators can be assigned to rate each pair of sentences as a paraphrase or not. The label given by each annotator can be represented as a matrix, with

each row and column representing a sentence pair and each cell containing the label given by a specific annotator. The Fleiss kappa formula can then be used to calculate the level of agreement among annotators, with higher scores indicating higher levels of agreement. The resulting score can provide insights into the quality of the dataset and the consistency of the annotations, which can inform decisions regarding the inclusion or exclusion of specific sentence pairs in the dataset. In Table 3 you can significance leve of dataset quality based on Fleiss Kappa.

**Table 3.** Significance table for Fleiss kappa.

| Kappa Score | Significance |
|---|---|
| <0 | No agreement |
| 0.01–0.20 | Slight agreement |
| 0.21–0.40 | Fair agreement |
| 0.41–0.60 | Moderate agreement |
| 0.61–0.80 | Substantial agreement |
| 0.81–0.99 | Almost perfect agreement |
| 1 | Perfect agreement |

*4.2. Experiment Scenario*

The paraphrase generation experiment was structured to examine the architectural abilities developed in the main domain of scientific paper sentences and the general domain. The model variations were used in all action factors in order to see the ability to produce paraphrased sentences in scientific paper sentences. As a result, 21 experimental scenarios can be produced (three StoPGEN architectures multiplied by seven action factor scenarios).

*4.3. Method Comparison*

4.3.1. UPSA

The main baseline of our study is the UPSA method [8]. This method is similar to the algorithm developed. However, they differ in the objective function, object selection scheme, and language domain used.

4.3.2. Variant Auto Encoder

In a previous study, the variant autoencoder method with the main LSTM architecture was used to generate the standard Twitter and Quora datasets [25]. This model uses the 300-dimensional LSTM trained with the non-parallel dataset. The mechanism is to maximize the loglikelihood in the inference process with the sentence variations choice obtained from the latent space feature.

4.3.3. Lagging Variant Auto Encoder

A simpler model from VAE was developed by increasing the sequence model learning ability, namely LagVAE [51], and reported to have better performance in the standard datasets used.

4.3.4. CGHM

CGHM [23] was developed using Metropolis–Hastings, which is a method for taking word space samples for sentence making. This method's results outperformed VAE in the case of latent space samples. Moreover, CGHM is an unsupervised paraphrasing technique that excels on standard datasets.

### 4.3.5. Modified UPSA

The UPSA baseline model was developed by changing the language model used. The UPSA language model is the Twitter and Quora domains. This replacement was built from scientific paper collections. The model includes two parts, namely, the candidate token selection and the probabilistic model for the objective function.

### 4.3.6. LSTM Encoder-Decoder

LSTM encoder-decoder was widely used in machine translation [21]. The code was rewritten by including the scientific papers' parallel sentences. This technique was included in the supervised method to examine the performance of the developed dataset.

### 4.3.7. Bi-Directional LSTM

Similar to the LSTM encoder-decoder, bi-directional is commonly used in generating interpreter machines [22]. This method adds an alternating learning scheme to sequentially arranged LSTM cells. It was only used in this study for the scientific paper's domain.

### 4.3.8. Transformer

The Transformer-based generation model utilizes the alignment token to see how closely related two text units are and subsequently aligns them into a matrix. Therefore, it can be trained as a spatial data model [19][24]. The method was originally developed for machine translation, and the most basic transformer architecture was used for paraphrase generation.

### *4.4. Matrix Evaluation*

### 4.4.1. BLEU Score

BLEU (bilingual evaluation understudy) is an evaluation method on the natural language generator that examines how close the machine output is to the sentence reference [34]. This matrix measures the proximity of the results to the customizable word segment. The supervised BLEU Score generation model is used to calculate one output based on the reference sentence. In generating unsupervised and stochastic models, the score can calculate all output results with reference sentences as well as the average and best results.

### 4.4.2. Rouge

Rouge (Recall-Oriented Understudy for Gisting Evaluation) is a software package for calculating. The generated sentences proximity with references that have variations includes ROUGE 1 (R1), ROUGE 2 (R2), and ROUGE L (RL) [52]. ROUGE 1 calculates the unigram overlap of the output sentences by reference, while ROUGE 2 calculates the overlapping bigrams. ROUGE L (longest common subsequence) calculates based on the cut of the longest segment in the compared sentence. This research uses rouge version 1.0.1.

## 5. Results

### *5.1. Dataset Agreement*

We conducted an agreement analysis among five observers in evaluating a dataset consisting of 3476 paraphrase or non-paraphrase classification data. Each observer provided a classification of paraphrase or non-paraphrase for each data point. Out of the total of 5720 data, 4975 data were classified as paraphrase by all observers, 745 data were classified as non-paraphrase by all observers, and 269 data had different classifications among the observers.

Fleiss kappa was used to assess the amount of agreement among the observers. The study resulted in a kappa score of 0.67, showing that the observers were in good agreement. This kappa number shows a substantial level of agreement at the 0.05 alpha level,

according to the Fleiss kappa significance table. Yet, according to the criteria in the Fleiss kappa significance table, this kappa number still falls in the "fair agreement" category.

*5.2. Quantitative Evaluation*

The StoPGEN model ability was tested with several architectural variants and all scenarios of action factors. Furthermore, its performance was compared to the public and the citation sentence datasets from scientific papers. The use of the compared model was grouped into two approaches, namely supervised and unsupervised. This study used the evaluation matrix BLEU, Rouge 1, Rouge 2, and Rouge L.

5.2.1. Generate the Quora and Twitter

The developed model was pre-tested with a standard dataset to examine its visibility. It can outperform both the supervised and unsupervised models. Furthermore, it significantly outperformed CGHM as the best-supervised model on the Quora dataset. Meanwhile, the Twitter dataset only exceeded by a few points. The unsupervised model and the basis for developing it, namely UPSA, can be outperformed in both datasets by numbers that are not too far apart. This is because UPSA and StoPGEN have the advantage of being quite adaptive in any domain. Table 4 showed the performances of the metodes on standard dataset.

**Table 4.** Performances on a standard dataset.

| Model | Twitter | | | Quora | | |
|---|---|---|---|---|---|---|
| | BLEU | Rouge 1 | Rouge 2 | BLEU | Rouge 1 | Rouge 2 |
| Supervised | | | | | | |
| VAE | 3.46 | 15.13 | 3.40 | 13.96 | 44.55 | 22.64 |
| LagVAE | 3.74 | 17.20 | 3.79 | 15.52 | 49.20 | 26.12 |
| CGHM | 5.32 | 19.96 | 5.44 | 15.73 | 48.73 | 26.12 |
| Unsupervised | | | | | | |
| UPSA | 5.30 | 19.96 | 5.44 | 18.21 | 59.51 | 32.63 |
| StoPGEN | **6.26** | **28.60** | **8.75** | **22.37** | **61.09** | **40.79** |

In table 4 it can be seen that StoPGEN evaluation results, which are given bold, show the best results

5.2.2. Generate Citation Sentences

StoPGEN was developed with three variations, each with a different nature in generating sentences.

1. StoPGEN$_1$: Generates a sentence with a random action by selecting a random word position.
2. StoPGEN$_2$: Generates sentences with random actions by selecting words sequentially based on their position order in the sentence.
3. StoPGEN$_3$: Generates sentences with random actions by selecting words sequentially based on their position order in the sentence and using language models to select candidate accessors.

The results showed that the 3rd variation, StoPGEN outperformed the previous two, indicating that the word selection order and language model filters can produce better sentences. The best scenario is normally distributing the probability for each action factor. Although this factor can be adjusted dynamically as needed, the qualitative results showed that dividing it evenly yields better performance. Table 5 shows the evaluation results of all StoPGEN variants with all action factor scenarios.

**Table 5.** Performances of StoPGEN on the citation dataset.

| Model | Action Factor | BLEU | Rouge 1 | Rouge 2 | Rouge L |
|---|---|---|---|---|---|
| StoPGEN$_1$ | A | 44.74 | 58.64 | 39.71 | 55.28 |
| | B | 46.23 | 60.34 | 40.63 | 56.73 |
| | C | 47.47 | 61.92 | 41.42 | 58.11 |
| | D | 48.78 | 63.56 | 42.25 | 59.54 |
| | E | 50.22 | 65.13 | 43.33 | 61.06 |
| | F | 51.59 | 66.71 | 44.22 | 62.43 |
| | G | 52.45 | 67.85 | 45.01 | 63.48 |
| StoPGEN$_2$ | A | 27.80 | 44.23 | 28.69 | 41.65 |
| | B | 35.07 | 50.61 | 32.33 | 47.48 |
| | C | 40.72 | 55.85 | 35.31 | 52.16 |
| | D | 45.20 | 60.06 | 38.02 | 56.19 |
| | E | 49.16 | 63.73 | 40.86 | 59.71 |
| | F | 52.47 | 66.98 | 43.59 | 62.70 |
| | G | 54.58 | 68.98 | 45.54 | 64.62 |
| StoPGEN$_3$ | A | 27.69 | 46.04 | 28.55 | 42.87 |
| | B | 35.13 | 52.83 | 32.86 | 49.23 |
| | C | 40.83 | 58.26 | 36.41 | 54.27 |
| | D | 45.77 | 62.68 | 39.69 | 58.42 |
| | E | 49.80 | 66.31 | 42.82 | 61.78 |
| | F | 53.36 | 69.44 | 45.65 | 64.69 |
| | **G** | **55.37** | **71.28** | **47.46** | **66.32** |

In Table 5 it can be seen that StoPGEN that use scenario G, which are given bold , show the best results

The citation dataset was tested using StopGEN$_3$ and compared with the others (Table 6). The supervised model was selected based on its general use in the machine translation domain. Meanwhile, the selected unsupervised model, namely UPSA and the modified UPSA, is the main baseline of this study. The UPSA was modified by replacing the source language used with the scientific paper domain. It also uses a language model as successor words and objective functions.

**Table 6.** Performance compared model on citation dataset.

| Model | BLEU | Rouge 1 | Rouge 2 | Rouge L |
|---|---|---|---|---|
| **Unsupervised** | | | | |
| StoPGEN$_3$ | 55.37 | 71.28 | 47.46 | 66.32 |
| UPSA | 21.20 | 45.93 | 15.43 | 41.55 |
| Modified UPSA | 33.81 | 51.25 | 26.67 | 45.94 |
| **Supervised** | | | | |
| LSTM encoder-decoder | 25.77 | 22.60 | 7.68 | 20.13 |
| bidirectional LSTM | 28.93 | 26.10 | 11.75 | 23.44 |
| Transformer | 18.91 | 20.70 | 7.83 | 18.46 |

We conduct more experiments to back up the findings of this study. The supervised model was trained with a pair of citation paraphrase sentences that were labeled true. All the models used a sequence-to-sequence approach. The experimental results showed that the best-supervised model to use is the Bidirectional LSTM [22], outperforming other machine translation models with a BLEU value of 28.93.

The original UPSA was directly used without changing the source. The results were worse than the UPSA, which was modified by replacing the source language with a corpus of scientific papers. However, StopPGEN can still outperform the modified UPSA. It also outperformed all models producing BLEU 55.37, Rouge 1 71.28, Rouge 2 47.46, and Rouge L 66.32.

*5.3. Qualitative Evaluation*

The qualitative evaluation was used to examine the StoPGEN generation results ability in the context of the resulting language. A survey was conducted on 30 readers with specifications in the computational linguistics field to see the acceptance. This study compared the acceptance of all variations of StoPGEN generation results and its comparison with other models. The results for each model are further detailed, showing direct examples.

5.3.1. Stochastic StoPGEN Results

The StoPGEN model developed can produce stochastic paraphrase sentences at any time of generation. As a result, the generated sentences can be customized in accordance with the tendency between lexical differences or semantic similarities. The results of generating sentences five times in this experiment are shown in Table 7.

**Table 7.** Stochastic results example.

| | |
|---|---|
| **Input** | We use pre-trained glove (pennington et. Al., 2014) embeddings for our purposes |
| **Target** | We use glove (pennington et. Al., 2014) for our word embeddings |
| Output 1 | Used trained glove (pennington et. Al., 2014) embeddings for implement through |
| Output 2 | Use pre-300 dimensional glove embeddings (pennington et. Al., 2014) word glove embeddings for our purposes |
| Output 3 | We use glove trained pretrained embedding (pennington et. Al., 2014) matrix for our through |
| Output 4 | Use trained glove word vectors trained glove word (pennington et. Al., 2014) embeddings for our purposes |
| Output 5 | We use pre-trained (pennington et. Al., 2014) embeddings for our purposes glove |

Table 7 shows the generation results with an actual value of 0.9 for the Meteor Score and 0.1 for the Pinc Score. The architecture used was StoPGEN, which qualitatively and quantitatively produced the highest value. The 10 generated results showed that all examples were quite acceptable, although not all were grammatically perfect. This variation was the objective target of the model developed from this study.

5.3.2. Human Acceptability

A survey was conducted by showing the generation results, and the reader was subsequently asked to select the acceptance status for each sentence (True/False). In this survey, the reader was assisted with a coloring aid as a sign of the selected action.

Table 8 shows the generation results of all StoPGEN variants. The results of StoPGEN1 obtained sentences with a higher lexical divergence because the target was selected randomly. It also allowed the same unit position of the text to get repeated action. Meanwhile, StoPGEN2 produced sentences with better semantic similarity, and the number of sentence units was relatively the same as the input because the target sentences were selected sequentially. StoPGEN3 obtained new sentences that were more readable than the others because a language model filtered replacement/addition text units.

**Table 8.** Paraphrase generation result of the proposed methods.

| Input | StoPGEN₁ | StoPGEN₂ | StoPGEN₃ |
|---|---|---|---|
| the model was implemented using tensorflow (abadi et. al., 2015) | the model was based this implemented transformer using tensorflow implemented (abadi et. al.,2015) | the implemented transformer model transformer was transformer pytorch using | the our implemented our model based was pytorch using tensorflow (abadi et. al., 2015) |

| | | tensorflow (abadi et. al., 2015) | |
|---|---|---|---|
| du et al. (2017) pioneered nn-based qg by adopting the seq2seq architecture (sutskever et at, 2014) | du et al. (2017) pioneered nn-based qg by adopting seq2seq architecture based (sutskever et at, 2014) | du al et al. (2017) pioneered nn-based qg by furthermore seq2seq implement (sutskever et at, 2014) | du et al. (2017) pioneered nn-based qg by using seq2seq models (sutskever et at, 2014) |
| we use the spanish-english ner corpus introduced in the 2018 calcs competition (aguilar et. al., 2018), which contains a total of 67,223 tweets with 808,663 tokens | details dimensional 200 the corpus introduced in the data calcs competition (aguilar et. al., 2018) contains a of 67,223 tweets with 808,663 scratch tokens | details employ pre spanish-english ner corpus introduced in the 2018 calcs competition (aguilar et. al., 2018) which contains was descent total of 67,223 tweets with poor tokens | we use the spanish-english corpus in 2018 calcs competition (aguilar et. al., 2018) been it contains by was work a total 67,223 preserving tweets with 808,663 tokens |

The survey results on the StoPGEN variant showed that the 3rd model had the highest value, with half the average acceptance value of all survey answers. This further supports the quantitative results that this model variation is the best. Meanwhile, the other two produced quite different values (see Table 9).

**Table 9.** Acceptability generation by all variants of StoPGEN.

| Model | Acceptability (with Action Label) |
|---|---|
| StoPGEN$_1$ | 26.45 |
| StoPGEN$_2$ | 27.09 |
| StoPGEN$_3$ | 50.96 |

In the second survey, we compare the paraphrase result of the baseline method, UPSA, and UPSA-modified approaches. (see Table 10). As a result, StoPGEN$_3$ had an acceptance value that outperformed the UPSA and UPSA baseline models, which was modified for the scientific paper domain. Furthermore, the basic UPSA had a low acceptance value because the source language used was not modified.

**Table 10.** Result of paraphrase generation by unsupervised methods.

| Input | UPSA | Modified UPSA | StoPGEN$_3$ |
|---|---|---|---|
| Tokenizing the output sentence: all words except special tokens are segmented by farasa (abdelali et al., 2016) and then tokenized with arabert tokenizer | The uk offers all words except special tokens of segmented by (abdelali et al., 2016) | The sentence all from initialize words from except user tokens are by farasa (abdelali et al., 2016) and then with tokenizer | Tokenizing the output sentence: all words tokens segmented farasa (abdelali et al., 2016) and then tokenized with arabert tokenizer |
| We use the spanish-english ner corpus introduced in the 2018 calcs competition (aguilar et al., 2018), which contains a total of 67,223 tweets with 808,663 tokens | We use the huge ner currency corpus introduced in the 2016 competition in (aguilar et al., 2018) | We use the spanishenglish corpus in 2018 calcs competition (aguilar et al., 2018) been it contains by was work a total 67,223 preserving tweets with 808,663 tokens | We the spanish-english ner corpus in the 2018 calcs competition (aguilar et al., 2018), which contains a total of 67,223 tweets with 808,663 tokens |
| The first one is heuristic rules such as treating identical words as the seed (artetxe et al., 2017), but this kind of method is restricted to languages sharing the alphabet | The one is heuristic rules such as treating identical words regarding the seed (artetxe et al., 2017) | First one transformer is transformer formation such bleu treating from senses identical from words as tialize the seed (artetxe et al., 2017) but partially work kind work of this method is this restricted to languages sharing the alphabet | The one heuristic rules such as the seed (artetxe et al., 2017), but this kind of method is restricted to languages sharing the alphabet |

Table 10 compares the UPSA outputs [8], where the source language was modified for the scientific papers domain and the best variant of the developed StoPGEN. The UPSA generation results showed a decrease in sentences because it produced many Out of Vocab (OOV). Furthermore, the results did not appear as there were language model limitations. Even though the output of the modified UPSA was not much OOV (lost), the readability was low. This is due to the different configurations of simulated annealing. Therefore, the StoPGEN output results are more acceptable with lexical variations and semantic similarities that are still maintained.

**Table 11.** Acceptability generation results by the compared model.

| Model | Acceptability (without Action Label) |
|---|---|
| UPSA | 16.80 |
| Modified UPSA | 26.40 |
| StoPGEN$_3$ | 50.80 |

Table 11 show paraphrase acceptability from all compared model, based on survey. StoPGEN$_3$ obtained the highest acceptance value even though the results differed from the previous survey. The acceptance rate of 50.80 indicates that more than half of the displayed results had received acceptance from readers. Values that are less than the first survey can be considered insignificant.

## 6. Conclusions

This study succeeded in developing a paraphrase generation model for the scientific paper domain, specifically citation sentences. Furthermore, it developed StoPGEN as a sentence generation model with three variants. The best results were shown by the variants that used sequential word selection strategies, with equal opportunities for substitution, insertion, and deletion actions. A probability-based language model was used to select a replacement token or sentence filler at the end of the generation.

The method in variant three has shown the best results with the action factor G. That action factor is a strategy for selecting text change actions (substitution, insertion, and deletion) with equal probability. In other action factor strategies, changes in probability have been analyzed but did not show better results.

The StoPGEN generation results were compared with other models in the domain. Our model outperformed the supervised one which is commonly used for machine translation sentence generation. This research compared the result with deep learning models such as LSTM, Bi-directional LSTM, variant autoencoder, and Transformer. The StoPGEN can outperform the results of these methods. We see the dataset as the factor, as we know that it was not big enough for a deep learning neural network base method. We see that the method we have developed is suitable for low language resources. It also outperformed the unsupervised UPSA and modified UPSA, which had its source language adapted for scientific papers.

Furthermore, the developed method performed better on public datasets than in other studies. StoPGEN had a superior performance for the Twitter and Quora datasets, with BLEU scores of 6.26 and 22.37, outperforming the other models.

In qualitative measurement by survey, the best StoPGEN variant had an acceptance value of 50.96. Meanwhile, the true value was 50.80 when the model was compared with others. From the output observations, StoPGEN can produce stochastic outputs while maintaining semantic similarities but with lexical differences.

## 7. Future Works

There are various drawbacks to this study. This study method is dependent on the corpus domain. According to qualitative evaluation data, the level of reader revenue remains around 50%. There are still grammatical mistakes in selecting successors in substitution and insertion actions.

Further research can be developed by considering several aspects, including lexical, syntactic, and semantic dimensions. In this study, the paraphrasing aspect only considers semantic similarity and language divergence. Further research can pay attention to fluency, diversity, and coherence. All aspects mentioned are expected to be implemented to make a better model. With a better model, the value of reader acceptance will be increased.

Future research could take several different directions that would build on the results and contributions of this work. Objective functions can be built by combining several properties of paraphrasing, such as similarity, fluency, diversity, coherence, and linguistic correctness. Utilizing the latest transformer-based language resources such as GPT-3, Bert, Roberta, and T5.

The latest transformer-based language resources can be used to improve paraphrase generation by providing more accurate and diverse paraphrases. These models can be fine-tuned on large-scale datasets to learn the relationship between different sentences and generate high-quality paraphrases. Additionally, incorporating syntax information into these models can further improve their performance. The use of unaligned pre-trained models can also help generate domain-specific paraphrases.

**Author Contributions:** Conceptualization, R.I., M.L.K.; Methodology, R.I.; Software, R.I; Formal analysis, R.I.; Data curation, M.L.K.; Writing—original draft, R.I.; Writing—review and editing, M.L.K.; Supervision, R.M. (Rinaldi Munir), M.L.K., R.M. (Rila Mandala) and D.H.W. All authors have read and agreed to the published version of the manuscript.

**Funding:** This research was funded by the Educational Fund Management Institution—Ministry of Finance, Indonesia. The APC was funded by the Educational Fund Management Institution—Ministry of Finance and Bandung Institute of Technology.

**Institutional Review Board Statement:** Not applicable.

**Informed Consent Statement:** Not applicable.

**Data Availability Statement:** The data presented in this study are available on request from the corresponding author.

**Acknowledgments:** We would like to express our appreciation to D.H.W., who contributed to this research before passing away. We are grateful for their dedication to leading this project and providing valuable insights during the initial stage of the research. Their presence is deeply missed, and their contribution will always be remembered and appreciated.

**Conflicts of Interest:** The authors declare no conflict of interest.

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
