# Peer review of "Generating Paraphrase Using Simulated Annealing for Citation Sentences"

_informatics, doi:10.3390/informatics10020034_

Round 1
Reviewer 1 Report
The authors of the paper propose a method for generating paraphrases of citation sentences using simulated annealing. The proposed method is based on the optimization of an objective function that measures the semantic similarity and lexical divergence between two pairs of sentences. The authors explain the formation of the objective function in the experiment section.
1- In the Introduction section, the authors should provide a comprehensive summary of the latest research work in the field, highlighting key findings and trends. This will help set the context for the paper and give the reader a good understanding of the state of the field.
Once the latest research work has been summarized, the authors should then identify gaps in the existing literature and clearly specify the contributions of their work. This will help the reader understand the unique value of the paper and the significance of the research being reported.
To improve the clarity and impact of the Introduction section, the authors should ensure that the summary of the latest research work is concise and relevant to the topic being studied. Additionally, the gaps and contributions should be clearly defined and well-articulated, making it easy for the reader to understand the significance of the research.
2- In the Related Work section, the authors should focus on the most recent research studies in the field to provide a comprehensive overview of the current state of the field. By building their work on the most recent studies, the authors can ensure that their work is up-to-date and relevant to the current body of knowledge.
To improve the impact of the Related Work section, the authors should carefully select the studies they include and ensure that they are relevant to their own research. Additionally, the authors should critically evaluate the strengths and limitations of the studies they include and provide an in-depth analysis of how their own work builds upon and extends the existing knowledge.
3- To improve the research method section, the authors could consider the following recommendations:
Incorporate additional techniques: The authors could consider incorporating additional techniques for measuring paraphrase, such as word embedding.
Experiment with combining techniques: The authors could experiment with different ways to combine semantic similarity and lexical divergence in the objective function to improve its effectiveness.
Increase the size and quality of the corpus: Increasing the size and quality of the corpus used could improve the results of the learning-based approaches.
Consider other evaluation metrics: The authors could consider using additional evaluation metrics, such as precision, recall, and F1 score, to provide a more comprehensive evaluation of the paraphrase generation results.
Further explanation of the objective function: The authors could provide a more detailed explanation of the objective function and how it was developed to provide a better understanding of its role in the generation model.
4- I would like to provide the following recommendations to improve the authors' work:
a. In Function 3, provide an explanation for the math notation used and consider using an equation instead of a function for better clarity.
b. Remove the coloring from Table 7 for improved readability.
c. Consider adding a diagram to illustrate the research methods and sequence of steps for better understanding.
d. Provide more detail on the limitations and future directions of the work to help other researchers benefit from the current study.
e. Clarify the dataset used for evaluating the performance of the proposed method and provide more information on the extraction and preprocessing of the data. This includes information on cleaning and filtering steps taken to remove noisy or irrelevant data, and normalization and standardization steps taken to ensure the data is suitable for analysis.
f. Provide a discussion of the impact of the preprocessing steps on the results of the proposed method, including a comparison of the results obtained using preprocessed data versus raw data.
g. In the discussion of the results, consider a more detailed discussion of other approaches to measuring paraphrases, including deep learning techniques, Siamese Neural Networks, and SVM models, and their strengths and weaknesses in comparison to the proposed method. This will provide valuable insight into the reliability and robustness of the proposed method.
Author Response
Dear Reviewer 1,
Thank you for giving advice and improvement to the paper we sent.
We have worked on all suggestions and revisions. We mark correction points on the document that is sent. The Review menu (MS. Word) displays all improvements.
Best Regard

Reviewer 2 Report
This paper experiments with a paraphrase generation approach in scientific literature, particularly in citations of other articles, using Simulated Annealing (SA). Taking into account semantic proximity measures based on word embeddings, and lexical divergence admissibility measures, an SA algorithm is featured to generate paraphrases that simultaneously maintain semantic proximity and lexical diversity. The idea of maintaining these two "forces", which tend to be opposed, seems to me to be a very interesting and promising idea for the generation of new phrases.
The work of this article is a small incremental, in relation to the UPSA method [8], bringing only the novelty of the variation in the objective function of the SA and in the language domain. However, the results reported by StoPGEN are quite better than those of UPSA.
From what I saw in the results, for this method to be applied in other text genres, not just scientific literature, it might be necessary to combine these two measures with something else, to ensure that the sentences generated are syntactically valid and written in a conventional way for the language being processed ("Acceptability"). It's just an idea to explore eventually in the future. I think that the level of syntactic errors in the generated sentences (Table 9, for StoPGEN3) is still unacceptable for practical applications. But as I said, this seems to me to be easily overcome.
The article is satisfactorily written, with some points less clear, some lapses and errors, which I will list in order below. For the article to be accepted, on my part, the points listed below must be treated/corrected carefully by the authors.
- P2/L49: "The citation used in this study was limited to only one paper" <== Unclear sentence. What do you mean exactly?
- P2/L60: In the last paragraph, before Section 2, you are quickly introducing a number of new concepts and unknown abbreviations. We have the references, but a small (one or two simple sentences) introductory description would be useful for the reader.
- P2/L83: "The construction of the citation paraphrase corpus in this study is a small contribution to capturing the existing potential." <== What do you mean exactly by "capturing the existing potential".
- P3/L101-103: "deep learning" ... "Besides the neural network basic architecture ..." <== Deep learning is far from basic!
- P3/L129: The sentence starting with "Moreover" is not well written. Must be corrected/completed or rephrased. Also, in the paper, you have a few cases with no space between the word and the citation number, like here "Transformer[16]". You should checkout all similar cases and correct them.
- P3/L133: "in the searching case of a very 133 large solution space [40]" ==> "in the solution search, on a very large dimension space [40]."
- P4/L157: "Simulate Annealing Algorithm" ==> "Simulated Annealing algorithm"
- P4: The Figure 1 caption does not end with a punctuation mark. This is observed on each caption throughout the paper. Must be rectified.
- P5/L195: In formula (2) what does n represent exactly? Is this a probability? How? Not clear!
- P5/L211: "The probability to choised for action was" ==> "The probability to choose for the action was"
- P5/L213: "from 3 available actions" <== What actions?
- P7/L272: "This value was selected based on the measurement of pre-experimental results." <== Can you provide further justification for such unbalanced weights? Maybe just one or two more sentences will clarify your decision in the reader’s mind.
- P7/L320: "Subsequently, each sentence is processed by clustering technique based on the text features ..." <== What do you mean exactly? Did you used any clustering algorithm for this task? Which one?
- P8/L343: "The main baseline of this study is" ==> "The main baseline of our study is". You can also change in other parts of the paper. It becomes much more clear for the reader if you refer directly to your paper/work.
- P9/L381: "4.3. Matric Evaluation" ==> "Matrix Evaluation" (?) Check this.
- P10/L431: "The best scenario evenly (uniformly) divides the opportunities for each action when viewed from the Action Factor." <== Unclear statement.
- P11/L438: What do you mean by “this study model? Is it StoPGEN3?
- P11/L446: "The supervised model was trained with a pair of citation paraphrase sentences" <== What was "trained"? Your approach is unsupervised! Please, explain this better.
- P12/L461: "are more detailed" ==> "are further detailed"
- P12/L468: "Table 5. Stochastic Results Example" ==> "Examples of stochastic paraphrase generations.". ATTENTION: Why does this table has two captions and numbers (5, and 6), one above and one below? The same problem is observed in the two tables that follow this one. This must be corrected!
- P13/L499-504. This paragraph is incomprehensible. Please, clarify this.
- P15/L529: "compared with other models in the scientific paper domain. This model outperformed" ==> "compared with other models in the domain. Our model outperformed"
- P15/L531: "It also outperformed the unsupervised UPSA and UPSA models" <== Why are you using two times "UPSA" in this sentence?
Author Response
Dear Reviewer 2,
Thank you for giving advice and improvement to the paper we sent.
We have worked on all suggestions and revisions. We mark correction points on the document that is sent. The Review menu (MS. Word) displays all improvements.
Best Regard

Round 2
Reviewer 1 Report
- Consider adding a comprehensive summary of the latest research work in the field to the Introduction section. This will help set the context for the paper and give the reader a good understanding of the state of the field.
- Highlight the research gaps in the current literature in the Introduction section.
- Clearly identify the main contribution or addition to the body of knowledge that the authors are making in the Introduction Section.
- Improve the quality of Figure 1 in the paper, as it is difficult to read and understand.
- I could not find paper [6], So Authors should add a section to the paper that explains the preprocessing steps taken to remove noisy or irrelevant data, and normalization and standardization steps taken to ensure the data is suitable for analysis.
- Discuss the limitations and future directions of the study in more detail, including potential avenues for future research and how the study's limitations might be addressed. Explore why the reader revenue remained at around 50% and what could be done to improve it. Additionally, provide more detail on how the latest transformer-based language resources could be used to improve paraphrase generation.
Author Response
Dear Reviewer 1,
Thank you for giving advice and improvement to the paper we sent in round 1.
We have worked on all suggestions and revisions. We mark correction points on the document that is sent. The Review menu (MS. Word) displays all improvements.
Best Regard

Reviewer 2 Report
1. You still have two tables 9 , one in line 563 and the other one in line 544, where above the table (line 43) you have caption saying it is Table 7. Something is wrong here and must be corrected.
2. In line 499 rewrite "StopGEN3" with 3 in subscript, has you have in the rest of the paper.
3. You still have jams of word+citation, like this one in line 35: "characteristics[4]" (line 35). You should ensure always a space between the word that precedes the citation.
4. Suggestion for a newly inserted sentence (line 493): "The best scenario is distributed normally the probability for each action Action Factor." ===> The best scenario is normally distributing the probability for each Action Factor.
Author Response
Dear Reviewer 2,
Thank you for giving advice and improvement to the paper we sent in round 1.
We have worked on all suggestions and revisions. We mark correction points on the document that is sent. The Review menu (MS. Word) displays all improvements.
Best Regard
